# TOWARDS LANGUAGE AGNOSTIC UNIVERSAL REPRESENTATIONS

## ABSTRACT

When a bilingual student learns to solve word problems in math, we expect the student to be able to solve these problem in both languages the student is fluent in, even if the math lessons were only taught in one language. However, current representations in machine learning are language dependent. In this work, we present a method to decouple the language from the problem by learning language agnostic representations and therefore allowing training a model in one language and applying to a different one in a zero shot fashion. We learn these representations by taking inspiration from linguistics, specifically the Universal Grammar hypothesis and learn universal latent representations that are language agnostic (Chomsky, 2014; Montague, 1970). We demonstrate the capabilities of these representations by showing that models trained on a single language using language agnostic representations achieve very similar accuracies in other languages.

## 1 INTRODUCTION

Anecdotally speaking, fluent bilingual speakers rarely face trouble translating a task learned in one language to another. For example, a bilingual speaker who is taught a math problem in English will trivially generalize to other known languages. Furthermore there is a large collection of evidence in linguistics arguing that although separate lexicons exist in multilingual speakers the core representations of concepts and theories are shared in memory (Altarriba, 1992; Mitchel, 2005; Bentin et al., 1985). The fundamental question we're interested in answering is on the learnability of these shared representations within a statistical framework.

We approached this problem from a linguistics perspective. Languages have vastly varying syntactic features and rules. *Linguistic Relativity* studies the impact of these syntactic variations on the formations of concepts and theories (Au, 1983). Within this framework of study, the two schools of thoughts are linguistic determinism and weak linguistic influence. *Linguistic determinism* argues that language entirely forms the range of cognitive processes, including the creation of various concepts, but is generally agreed to be false (Hoijer, 1954; Au, 1983). Although there exists some weak linguistic influence, it is by no means fundamental (Ahearn, 2016). The superfluous nature of syntactic variations across languages brings forward the argument of *principles and parameters* (PnP) which hypothesizes the existence of a small distributed parameter representation that captures the syntactic variance between languages denoted by parameters (e.g. head-first or head-final syntax), as well as common principles shared across all languages (Culicover, 1997). *Universal Grammar* (UG) is the study of principles and the parameters that are universal across languages (Montague, 1970).

The ability to learn these universalities would allow us to learn representations of language that are fundamentally agnostic of the specific language itself. Doing so would allow us to learn a task in one language and reap the benefits of all other languages without needing multilingual datasets. We take inspiration from the UG hypothesis and learn latent representations that are language agnostic which allow us to solve downstream problems in new languages without the need of any language-specific training data. We do not make any claims about the Universal Grammar hypothesis, but simply take inspiration from it.

## 2 RELATED WORK

Our work attempts to unite universal (task agnostic) representations with multilingual (language agnostic) representations (Peters et al., 2018; McCann et al., 2017). The recent trend in universal representations has been moving away from context-less unsupervised word embeddings to context-rich representations. Deep contextualized word representations (ELMo) trains an unsupervised language model on a large corpus of data and applies it to a large set of auxiliary tasks (Peters et al., 2018). These unsupervised representations boosted the performance of models on a wide array of tasks. Along the same lines McCann et al. (2017) showed the power of using latent representations of translation models as features across other non-translation tasks. In general, initializing models with pre-trained language models shows promise against the standard initialization with word embeddings. Even further, Radford et al. (2017) show that an unsupervised language model trained on a large corpus will contain a neuron that strongly correlates with sentiment without ever training on a sentiment task implying that unsupervised language models maybe picking up informative and structured signals.

In the field of multilingual representations, a fair bit of work has been done on multilingual word embeddings. Ammar et al. (2016) explored the possibility of training massive amounts of word embeddings utilizing either parallel data or bilingual dictionaries via the SkipGram paradigm. Later on an unsupervised approach to multilingual word representations was proposed by Chen & Cardie (2018) which utilized an adversarial training regimen to place word embeddings into a shared latent space. Although word embeddings show great utility, they fall behind methods which exploit sentence structure as well as words. Less work has been done on multilingual sentence representations. Most notably both Schwenk & Douze (2017) and Artetxe et al. (2017) propose a way to learn multilingual sentence representation through a translation task.

We train downstream models using language agnostic universal representations on a set of tasks and show the ability for the downstream models to generalize to languages that we did not train on.

## 3 OPTIMIZATION PROBLEM

Statistical language models approximate the probability distribution of a series of words by predicting the next word given a sequence of previous words.

$$p(w_0, ..., w_n) = \prod_{i=1}^{n} p(w_i \mid w_0, ..., w_{i-1})$$

where $w_i$ are indices representing words in an arbitrary vocabulary.

Learning grammar is equivalent to language modeling, as the support of $p$ will represent the set of all grammatically correct sentences. Furthermore, let $j_\alpha$ represent a particular language. Let $p_{j_\alpha}(\cdot)$ represent the language model for the $j_\alpha^{\text{th}}$ language and $w^{j_\alpha}$ represents a word from the $j_\alpha^{\text{th}}$ language. Let $k_{j_\alpha}$ represent a distributed representation of a specific language along the lines of the PnP argument (Culicover, 1997). UG, through the lens of statistical language modeling, hypothesizes the existence of a factorization of $p_{j_\alpha}(\cdot)$ containing a language agnostic segment. The factorization used throughout this paper is the following ($\circ$ denotes function composition):

$$b = u \circ e_{j_\alpha}(w_0^{j_\alpha}, ..., w_i^{j_\alpha}) \tag{1}$$

$$p_{j_\alpha}(w_i \mid w_0, ..., w_{i-1}) = e_{j_\alpha}^{-1} \circ h(b, k_{j_\alpha}) \tag{2}$$

$$s.t. \quad d(\mathbf{p}(b \mid j_\alpha) \mid\mid \mathbf{p}(b \mid j_\beta)) \leq \epsilon \tag{3}$$

The distribution matching constraint $d$, insures that the representations across languages are common as hypothesized by the UG argument.

Function $e_{j_\alpha} : \mathbb{N}^i \to \mathbb{R}^{i \times d}$ is a language specific function which takes an ordered set of integers representing tokens (token meaning $w_i^{j_\alpha}$) and outputs a vector of size $d$ per token. Function $u : \mathbb{R}^{i \times d} \to \mathbb{R}^{i \times d}$ takes the language specific representation and attempts to embed into a language agnostic representation. Function $h : (\mathbb{R}^{i \times d}, \mathbb{R}^f) \to \mathbb{R}^{i \times d}$ takes the universal representation as well

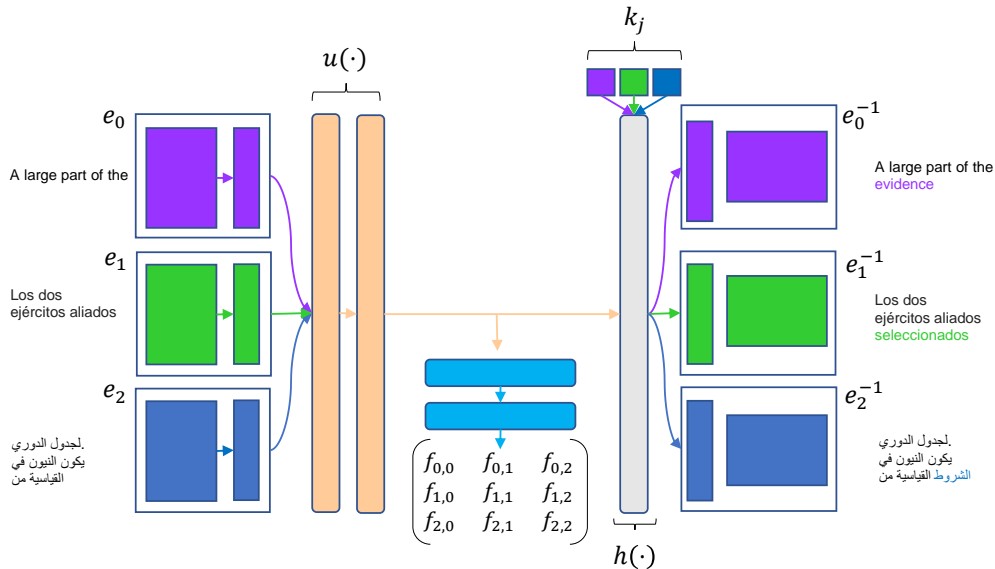

Figure 1: Architecture of UG-WGAN. The amount of languages can be trivially increased by increasing the number of language agnostic segments $k_{j_\alpha}$ and $e_{j_\alpha}$.

as a distributed representation of the language of size $f$ and returns a language specific decoded representation. $e_{j_\alpha}^{-1}$ maps our decoded representation back to the token space.

For the purposes of distribution matching we utilize the GAN framework. Following recent successes we use Wasserstein-1 as our distance function $d$ (Arjovsky et al., 2017).

Given two languages $j_\alpha$ and $j_\beta$ the distribution of the universal representations should be within $\epsilon$ with respect to the $W_1$ of each other. Using the Kantarovich-Rubenstein duality (Arjovsky et al., 2017; Villani, 2008) we define

$$d(\mathbf{p}(b \mid j_\alpha) \mid\mid \mathbf{p}(b \mid j_\beta)) = \sup_{||f_{\alpha,\beta}||_L \leq 1} \mathbb{E}_{x \sim \mathbf{p}(b|j_\alpha)} [f_{\alpha,\beta}(x)] - \mathbb{E}_{x \sim \mathbf{p}(b|j_\beta)} [f_{\alpha,\beta}(x)] \qquad (4)$$

where $L$ is the Lipschitz constant of $f$. Throughout this paper we satisfy the Lipschitz constraint by clamping the parameters to a compact space, as done in the original WGAN paper (Arjovsky et al., 2017). Therefore the complete loss function for $m$ languages each containing $N$ documents becomes:

$$\max_\theta \sum_{\alpha=1}^{m} \sum_{i=0}^{N} \log p_{j_\alpha}(w_{i,0}^{j_\alpha}, ..., w_{i,n}^{j_\alpha}; \theta) - \frac{\lambda}{m^2} \sum_{\alpha=1}^{m} \sum_{\beta=1}^{m} d(\mathbf{p}(b \mid j_\alpha) \mid\mid \mathbf{p}(b \mid j_\beta))$$

$\lambda$ is a scaling factor for the distribution constraint loss.

## 4 UG-WGAN

Our specific implementation of our factorization and optimization problem we denote as UG-WGAN.

### 4.1 ARCHITECTURE

Each function described in the previous section we implement using neural networks. For $e_{j_\alpha}$ in equation 1 we use a language specific embedding table followed by a LSTM (Hochreiter & Schmidhuber, 1997). Function $u$ in equation 1 is simply stacked LSTM's. Function $h$ in equation 2 takes

input from $u$ as well as a PnP representation of the language via an embedding table. Calculating the real inverse of $e_{j_\alpha}^{-1}$ is non trivial therefore we use another language specific LSTM whose outputs we multiply by the transpose of the embedding table of $e_{j_\alpha}$ to obtain token probabilities. For regularization we utilized standard dropout after the embedding layers and layer-wise locked dropout after each LSTM's layer (Srivastava et al., 2014; Gal & Ghahramani, 2016).

The critic, adopting the terminology from Arjovsky et al. (2017), takes the input from $u$, feeds it through a stacked LSTM, aggregates the hidden states using linear sequence attention as described in DrQA (Chen et al., 2017). Once we have the aggregated state we map to a $m \times m$ matrix from where we can compute the total Wasserstein loss. A Batch Normalization layer is appended to the end of the critic (Ioffe & Szegedy, 2015). The $\alpha, \beta$th index in the matrix correspond to the function output of $f$ in calculating $W_1(\mathbf{p}(b \mid j_\alpha) \mid\mid \mathbf{p}(b \mid j_\beta))$.

## 4.2 TRAINING

We trained UG-WGAN with a variety of languages depending on the downstream task. For each language we utilized the respective Wikipedia dump. From the wikipedia dump we extract all pages using the wiki2text[1] utility and build language specific vocabularies consisting of 16k BPE tokens (Sennrich et al., 2015). During each batch we uniform sample random documents from our set of languages which are approximately the same length, therefore a batch will be mixed with respect to language. We train our language model via BPTT where the truncation length progressively grows from 15 to 50 throughout training. The critic is updated 10 times for every update of the language model. We trained each language model for 14 days on a NVidia Titan X. For each language model we would do a sweep over $\lambda$, but in general we have found that $\lambda = 0.1$ works sufficiently well for minimizing both perplexity and Wasserstein distance.

## 4.3 EXPLORATION

A couple of interesting questions arise from the described training procedure. Is the distribution matching constraint necessary or will simple joint language model training exhibit the properties we're interested in? Can this optimization process fundamentally learn individual languages grammar while being constrained by a universal channel? What commonalities between languages can we learn and are they informative enough to be exploited?

We can test out the usefulness of the distribution matching constraint by running an ablation study on the $\lambda$ hyper-parameter. We trained UG-WGAN on English, Spanish and Arabic wikidumps following the procedure described above. We kept all the hyper-parameters consistent apart for augmenting $\lambda$ from 0 to 10. The results are shown in Figure 2. Without any weight on the distribution matching term the critic trivially learns to separate the various languages and no further training reduces the wasserstein distance. The joint language model internally learns individual language models who are partitioned in the latent space. We can see this by running a t-SNE plot on the universal ($u(\cdot)$) representation of our model and seeing existence of clusters of the same language as we did in Figure 3 (Maaten & Hinton, 2008). An universal model satisfying the distribution matching constrain would mix all languages uniformly within it's latent space.

To test the universality of UG-WGAN representations we will apply them to a set of orthogonal NLP tasks. We will leave the discussion on the learnability of grammar to the Discussion section of this paper.

## 5 EXPERIMENTS

By introducing a universal channel in our language model we reduced a representations dependence on a single language. Therefore we can utilize an arbitrary set of languages in training an auxiliary task over UG encodings. For example we can train a downstream model only on one languages data and transfer the model trivially to any other language that UG-WGAN was trained on.

---

[1] https://github.com/rspeer/wiki2text

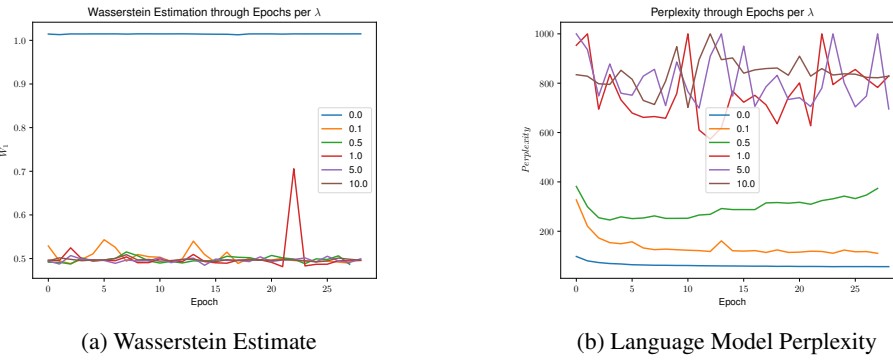

(a) Wasserstein Estimate

(b) Language Model Perplexity

Figure 2: Ablation study of $\lambda$. Both Wasserstein and Perplexity estimates were done on a held out test set of documents.

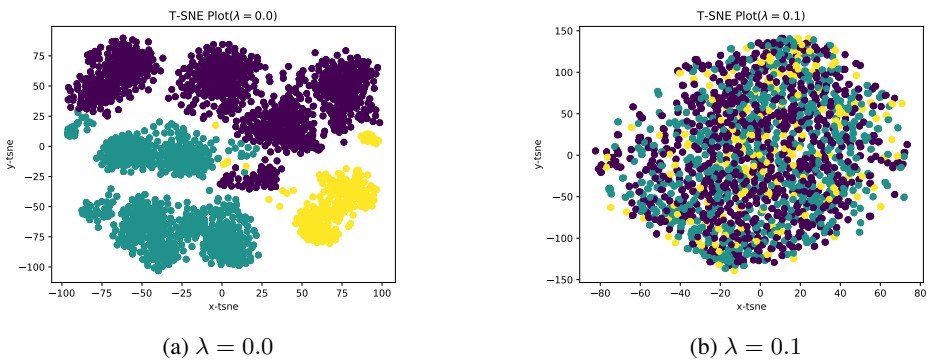

(a) $\lambda = 0.0$

(b) $\lambda = 0.1$

Figure 3: T-SNE Visualization of $u(\cdot)$. Same colored dots represent the same language.

### 5.1 SENTIMENT ANALYSIS

To test the universality of UG-WGAN representation we first trained UG-WGAN in English, Chinese and German following the procedure described in Section 4. The embedding size of the table was 300 and the internal LSTM hidden size was 512. A dropout rate of $0.1$ was used and trained with the ADAM optimization method (Kingma & Ba, 2014). Since we are interested in the zero-shot capabilities of our representation, we trained our sentiment analysis model only on the English IMDB Large Movie Review dataset and tested it on the Chinese ChnSentiCorp dataset and German SB-10K (Maas et al., 2011; Tan & Zhang, 2008). We binarize the label's for all the datasets.

Our sentiment analysis model ran a bi-directional LSTM on top of fixed UG representations from where we took the last hidden state and computed a logistic regression. This was trained using standard SGD with momentum.

| Method | IMDB | ChnSentiCorp | SB-10K |
|---|---|---|---|
| NMT + Logistic (Schwenk & Douze, 2017) | 12.44% | 20.12% | 22.92% |
| FullUnlabeledBow (Maas et al., 2011) | 11.11% | * | * |
| NB-SVM TRIGRAM (Mesnil et al., 2014) | 8.54% | 18.20% | 19.40% |
| **UG-WGAN** $\lambda = 0.1$ **+ Logistic (Ours)** | 8.01% | 15.40% | 17.32% |
| UG-WGAN $\lambda = 0.0$ + Logistic (Ours) | 7.80% | 53.00% | 49.38% |
| Sentiment Neuron Radford et al. (2017) | 7.70% | * | * |
| SA-LSTM (Dai & Le, 2015) | 7.24% | * | * |

Table 1: Zero-shot capability of UG and OpenNMT representation from English training. For all other methods we trained on the available training data. Table shows error of sentiment model.

We also compare against encodings learned as a by-product of multi-encoder and decoder neural machine translation as a baseline (Klein et al., 2017). We see that UG representations are useful in situations when there is a lack of data in an specific language. The language agnostics properties of UG embeddings allows us to do successful zero-shot learning without needing any parallel corpus, furthermore the ability to generalize from language modeling to sentiment attests for the universal properties of these representations. Although we aren't able to improve over the state of the art in a single language we are able to learn a model that does surprisingly well on a set of languages without multilingual data.

## 5.2 NLI

A natural language inference task consists of two sentences; a premise and a hypothesis which are either contradictions, entailments or neutral. Learning a NLI task takes a certain nuanced understanding of language. Therefore it is of interest whether or not UG-WGAN captures the necessary linguistic features. For this task we use the Stanford NLI (sNLI) dataset as our training data in English (Bowman et al., 2015). To test the zero-shot learning capabilities we created a russian sNLI test set by random sampling 400 sNLI test samples and having a native russian speaker translate both premise and hypothesis to russian. The label was kept the same.

For this experiment we trained UG-WGAN on the English and Russian language following the procedure described in Section 4. We kept the hyper-parameters equivalent to the Sentiment Analysis experiment. All of the NLI model tested were run over the fixed UG embeddings. We trained two different models from literature, Densely-Connected Recurrent and Co-Attentive Network by Kim et al. (2018) and Multiway Attention Network by Tan et al. (2018). Please refer to this papers for further implementation details.

| Method | sNLI(en) | sNLI (ru) |
|---|---|---|
| Densely-Connected Recurrent and Co-Attentive Network Ensemble (Kim et al., 2018) | **9.90%** | * |
| **UG-WGAN ($\lambda = 0.1$) + Densely-Connected Recurrent and Co-Attentive Network (Kim et al., 2018)** | 12.25% | **21.00%** |
| UG-WGAN ($\lambda = 0.1$) + Multiway Attention Network (Tan et al., 2018) | 21.50% | 34.25% |
| UG-WGAN ($\lambda = 0.0$) + Multiway Attention Network (Tan et al., 2018) | 13.50% | 65.25% |
| UG-WGAN ($\lambda = 0.0$) + Densely-Connected Recurrent and Co-Attentive Network (Kim et al., 2018) | 11.50% | 68.25% |
| Unlexicalized features + Unigram + Bigram features (Bowman et al., 2015) | 21.80% | 55.00% |

Table 2: Error in terms of accuracy for the following methods. For *Unlexicalized features + Unigram + Bigram features* we trained on 200 out of the 400 Russian samples and tested on the other 200 as a baseline.

UG representations contain enough information to non-trivially generalize the NLI task to unseen languages. That being said, we do see a relatively large drop in performance moving across languages which hints that either our calculation of the Wasserstein distance may not be sufficiently accurate or the universal representations are biased toward specific languages or tasks.

One hypothesis might be that as we increase $\lambda$ the cross lingual generalization gap (difference in test error on a task across languages) will vanish. To test this hypothesis we conducted the same experiment where UG-WGAN was trained with a $\lambda$ ranging from 0 to 10. From each of the experiments we picked the model epoch which showed the best perplexity. The NLI specific model was the Densely-Connected Recurrent and Co-Attentive Network.

Increasing $\lambda$ doesn't seem to have a significant impact on the generalization gap but has a large impact on test error. Our hypothesis is that a large $\lambda$ doesn't provide the model with enough freedom to learn useful representations since the optimizations focus would largely be on minimizing the Wasserstein distance, while a small $\lambda$ permits this freedom. One reason we might be seeing this generalization gap might be due to the way we satisfy the Lipschitz constraint. It's been shown that there are better constraints than clipping parameters to a compact space such as a gradient penalty (Gulrajani et al., 2017). This is a future direction that can be explored.

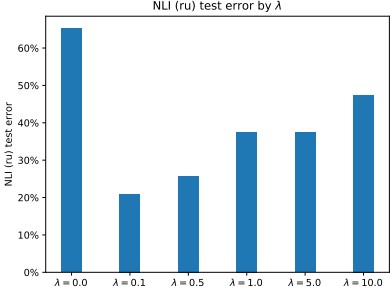
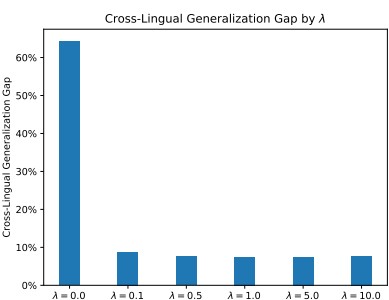

Figure 4: Cross-Lingual Generalization gap and performance

## 6 DISCUSSION

Universal Grammar also comments on the learnability of grammar, stating that statistical information alone is not enough to learn grammar and some form of native language faculty must exist, sometimes titled the poverty of stimulus (POS) argument (Chomsky, 2010; Lewis & Elman, 2001). The goal of our paper is not to make a statement on the Universal Grammar hypothesis. But from a machine learning perspective, we're interested in extracting informative features. That being said it is of interest to what extent language models capture grammar and furthermore the extent to which models trained with our objective learn grammar.

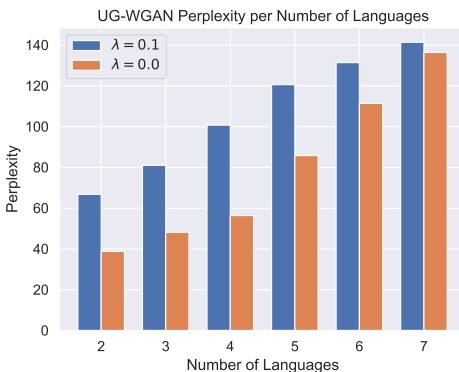
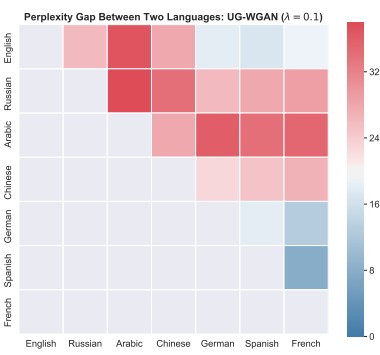

Figure 5: The figure on the left shows perplexity calculations on a held out test set for UG-WGAN trained on a varying number of languages. The figure on the right shows the perplexity gap between two languages trained with UG-WGAN $\lambda = 0.1$ and UG-WGAN $\lambda = 0.0$.

One way to measure universality is by studying perplexity of our multi-lingual language model as we increase the number of languages. To do so we trained 6 UG-WGAN models on the following languages: English, Russian, Arabic, Chinese, German, Spanish, French. We maintain the same procedure as described above. The hidden size of the language model was increased to 1024 with 16K BPE tokens being used. The first model was trained on English Russian, second was trained on English Russian Arabic and so on. For Arabic we still trained from left to right even though naturally the language is read from right to left. We report the results in Figure 5. As we increase the number of languages the perplexity gap between constrained and unconstrained UG-WGAN ($\lambda = 0.0$) decreases which implies while controlling capacity, our constrained (universal $\lambda = 0.1$) language model, models language (almost) as well as jointly trained language models with no universal constraints ($\lambda = 0.0$).

Furthermore, the heatmap in Figure 5 shows the perplexity gap of UG-WGAN trained on any combination of 2 languages from our set of 7. We can treat the perplexities as a loose measure of distance

| | $\lambda = 0.0$ | $\lambda = 0.1$ |
|---|---|---|
| en | earth's oxide is a monopoly that occurs towing of the carbon-booed trunks, resulting in a beam containing of oxygen through the soil, salt, warm waters, and the different proteins. | the practice of epimatic behaviours may be required in many ways of all non-traditional entities. |
| | the groove and the products are numeric because they are called "pressibility" (ms) nutrients containing specific different principles that are available from the root of their family, including a wide variety of molecular and biochemical elements. | a state line is a self-government environment for statistical cooperation, which is affected by the monks of canada, the east midland of the united kingdom. |
| | however, compared to the listing of special definitions, it has evolved to be congruent with structural introductions, allowing to form the chemical form. | the vernacular concept of physical law is not as an objection (the whis) but as a universal school. |
| es | la revista ms reciente vari el manuscrito originalmente por primera vez en la revista publicada en 1994. | en el municipio real se localiza al mar del norte y su entorno en escajros alto, con mayor variedad de cclica poblacin en forma de cerca de 1070 km2. |
| | de hecho la primera cancin de "blebe cantas", pahka zanjiwtryinvined cot de entre clases de fanticas, apareci en el ornitlogo sello triusion, jr., en la famosa publicacin playboy de john allen. | fue el ltimo habitantes de suecia, con tres hijos, atasaurus y aminkinano (nuestra). |
| | The names of large predators in charlesosaurus include bird turtles hibernated by aerial fighters and ignored fish. | jaime en veracruz fue llamado papa del conde mayor de valdechio, hijo de diego de ziga. |

Table 3: Example of samples from UG-WGAN with $\lambda = 0.0$ and $\lambda = 0.1$

between two languages. From the heatmap we see that English and Spanish are the most similar languages while Russian and Arabic are the most different, which aligns with intuition.

We see from Figure 2 that perplexity worsens proportional to $\lambda$. We explore the differences by sampling sentences from an unconstrained language model and $\lambda = 0.1$ language model trained towards English and Spanish in Table 3. In general there is a very small difference between a language model trained with our objective and one without. The constrained model tends to make more gender mistakes and mistakes due to Plural-Singular Form in Spanish. In English we saw virtually no fundamental differences between the language models. One explanation of this phenomena comes from the autonomy of syntax argument, which argues that semantics have no weight on syntax (Higginbotham, 1987). Our hypothesis is that both models learn syntax well, but the models with better perplexity generate sentences with *better* or *clearer* semantic meaning. Although completely learning grammar from statistical signals might be improbable, we can still extract useful information.

## 7   CONCLUSION

In this paper we introduced an unsupervised approach toward learning language agnostic universal representations by taking inspiration from the Universal Grammar hypothesis. We showed that we can use these representations to learn tasks in one language and automatically transfer them to others with no additional training. Furthermore we studied the importance of the Wasserstein constraint through the $\lambda$ hyper-parameter. And lastly we explored the difference between a standard multi-lingual language model and UG-WGAN by studying the generated outputs of the respective language models as well as the perplexity gap growth with respect to the number of languages.

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
