# OpenReview forum: "Towards Language Agnostic Universal Representations"
_ICLR.cc/2019/Conference_

### Official Review · AnonReviewer3 · 2018-10-29
**Good method and results overall, with a few questions on analysis**

**Rating:** 6
**Confidence:** 3

**Review:**

This paper introduced a GAN-based method to learn language universal representations without parallel data. The model architecture is analogous to an autoencoder. The encoder is a compound of language-universal mapper plus a language-specific LSTM. For decoding, another language-universal module first map language-universal representation back to language-specific embedding space, then another LSTM decoder generates the original sentence. The authors used GAN to encourage intermediate representation to be language-universal. The authors tested the proposed method on zero-shot semantic analysis and NLI tasks and showed nice results.

Overall the proposed method is novel and nice, and experiment results are good. On both tasks the proposed method performs better than NMT methods on target languages while still achieving competitive performance on source languages. The paper is also clearly written and could be useful for future research on multilingual transfer.

My main complaint is around Figure 5, Table 3, and the corresponding analysis.
1. In Figure 5, does it make more sense to show the perplexity of a standard LM. That is, train 7 independent LMs and report averaged perplexity. My concern is that, even with \lambda=0.0, the model still have modules u and h that are shared across languages, and therefore I'm not sure if it implies "representative power of UG-WGAN grows as we increase the number of languages". It could be that the language-universal impose more constraints to model all languages, so the two variation (\lambda=0.0 or 0.1) come closer to each other.

2. In Figure 3, the perplexity difference is huge when number of languages is 2. In Table 3, however, the authors show no fundamental differences between the English and Spanish language models. I feel the two arguments contradict to each other. Is it because of the language pairs are different? The authors should provide more explanation on that.

Minor:
1. Equation 1 and 2 in page 2. Are they both compound functions? Why the first one use \circ and the second one use parenthesis?

---

> ### Author Response · Authors · 2018-11-06
> **RE: ICLR 2019 Conference Paper24 AnonReviewer3**
>
> Thank you for your detailed review. We'd like to comment on both of the points you've made.
>               1. We've internally debated prior to posting the paper whether or not to show perplexities of a standard LM vs the jointly trained language model. We decided to display the jointly trained language model with \lambda=0.0 due to the fact that we wanted to control over representative capacity of our language models. If we had 7 independent LM's, we would have to choose hyper-parameters in such a way that representative capacity of all 7 summed up to one UG-WGAN (in order to make the comparison informative). This is a non-trivial task. Furthermore in ~Figure 3, in the t-SNE plot of \lambda=0.0 we see that even the universal portion of UG-WGAN becomes language specific without a distribution constraint, therefore we can think of UG-WGAN \lambda=0.0 with $n$ languages as $n$ different LM's with a shared constraint of representative power. We believe our statement "representative power of UG-WGAN grows as we increase the number of languages" is rather ambiguous. What we were trying to communicate is as we increase the number of languages the perplexity gap between constrained and unconstrained UG-WGAN (\lambda=0.0) decreases which implies while controlling capacity, our constrained (universal \lambda=0.1) language model, models language (almost) as well as jointly trained language models with no universal constraints (\lambda=0.0). We have updated our paper clarifying our ambiguity.
>               2. This is a great point. One of the reasons that the perplexity difference is large in Figure 2 (which is the figure I think you're referring to, not 3) is because of the differences in language pairs, as you mentioned. But even so it's of interest to us that we see no fundamental difference between the English and Spanish language models. Our explanation of this follows as such. Language modeling attempts to first model syntax and then semantics. From a syntax perspective both the universal and standard models more or less learn the syntax, and can construct sentences which grammatically look and sound correct without having much semantic meaning. Semantic meaning is not necessary in order to construct syntactically correct sentences (i.e. autonomy of syntax hypothesis). We believe although there is no discernible difference between the sentences generated syntactically, the models with a smaller perplexity generate sentences which contain more of a semantic meaning. We have updated our paper to include this hypothesis.
>
> We've also fixed the \circ issue in the recent draft.
>
> Thank you once again for reading our paper and giving us a detailed reply.
> Looking forward to your reply

---

> > ### Comment · AnonReviewer3 · 2018-11-17
> > **Thank you; a suggestion on point 2, and I agreed with Reviewer1**
> >
> > On point 2, I actually meant Figure 5 where you showed perplexity for 2-7 pairs. If I understand correctly, you picked English and Russian for x=2, and the perplexity difference was big -- sorry for the confusion.
> >
> > Thanks for giving explanation, but I would like to see perplexity difference between Spanish and English as well, and I would expect they are not as big as that between Russian and English. It would be nice to have some more discussion on that as well.
> >
> > Regarding Reviewer1's complaint, I agree that some sentences need to be rephrased to make the claim better calibrated with your experiments. As Reviewer1 mentioned, the claim of learning universal "grammar" may confuse the reader that the model is able to learn "basic syntactic rules", but actually the model is just learning "hidden representations" of languages. I'd like the authors to follow Reviewer1's suggestions and make changes.

---

> > > ### Author Response · Authors · 2018-11-24
> > > **RE: AnonReviewer3**
> > >
> > > Thank you for your reply.
> > >
> > > We agree that seeing perplexity gaps across all combinations of two languages from our 7 would be a useful graphic. We ran the experiment you've mentioned prior to writing the paper to see if the various perplexities align with our intuition about languages. We've included a heat-map in Figure 5 showing this data. We see that perplexity gap between Russian and English is greater than Spanish and English, as you predicted in your previous comment. We've also included a short summary and description of the graphic.
> > >
> > > We also agree with the AnonReviewer1's criticism. Our updated revision removes/augments any statements about us "formalizing UG" or anything of that similar fashion.
> > >
> > > Please let us know if you have any further concerns.
> > > Thank you.

---

### Official Review · AnonReviewer2 · 2018-11-02
**The writing is unclear...**

**Rating:** 4
**Confidence:** 4

**Review:**

This paper proposes the idea of language agnostic representation which could potentially provide zero shot solution if the downstream task is trained using another language. The solution uses linguistic features from every sentence, trains language model for multiple languages simultaneously, and matches distribution by using Wasserstein distance measure.

pros:
The motivation of this paper is clear.
The method proposed looks reasonable.
The experimental results also make sense.

cons:

Key technical parts are not clear. The description of the training method is vague, e.g., the author(s) mentioned 'we utilized dropout and locked dropout where appropriate'. What does 'appropriate' mean? The training procedure was described in only few sentences. For example, it is not clear to me if a batch is fully random, or a batch consists of same number of sentences from each language, or a batch consists of same number of sentences from two languages, and how you train the WGAN. It is a bit surprising to me that different lambda gives similar performance.

The writing of the paper is not clear. Here are some of the reasons:
1. The last paragraph in Section 2 does not fit into 'related work' section at all, instead, it is almost a repetition of the last paragraph in Section 1.
2. The notations in Section 3 are very inconsistent. Just to name a few: the input dimension of function $e_j$ defined in the last paragraph in page 2 is not consistent with (1); the '$\circle$' operation in (1) is not explained (although I can guess what it means); the $j_alpha, j_beta$ are not consistent with the $j^{th}$ language; in the last equation in page 3, the summation should be from 1 to m (instead of 0 to m) if there are m languages, and the superscript in $w$ is not defined.
3. Key references missing, for example: there is no reference when deriving (4) using the 'Kantarovich-Rubenstein' duality.
4. The organization for section 4 is not clear. The first sentence is quite confusing, and the content is a mixture of architecture design, training details, and experimental settings. Instead, one should separate these contents and address each of them.
5. At the beginning of section 5.1, the hypothesis in the sentence 'to test this hypothesis' actually refers to the last paragraph in section 4. Figure 4 should be referred to in the last paragraph in section 5. 'english', 'german', 'chinese' should be 'English', 'German', 'Chinese'.

---

> ### Author Response · Authors · 2018-11-06
> **RE: ICLR 2019 Conference Paper24 AnonReviewer2**
>
> Thank you for taking the time to read our paper.
>
> We've taken your comments on our writing to heart and have updated our draft taking all your points into account. Please let us know if you have further comments on our writing.
>
> Thank you once again for you review.

---

### Official Review · AnonReviewer1 · 2018-11-02
**Very good work on learning language-agnostic embeddings but makes bold claims that are not verified**

**Rating:** 5
**Confidence:** 4

**Review:**

This paper starts with the bold aim of extracting Montague's universal grammar from multiple languages. In order to do so, the authors train multiple language models where each LM is explicitly factorized into language-specific and language-independent representations. The authors then apply the GAN framework to the language-independent parts to enforce all languages to share the same latent space. The claim here is that the language independent parameters capture the essence of universal grammar. The authors show that their framework enables effective zero-shot learning of tasks over new languages (for example sentiment classifier learned on top of English data generalize to Chinese when trained on the universal grammar embeddings).

The paper is overall well written and the experimental results are convincing.

The gripe, however, I have is that this paper makes the claims that go too far without evaluating them. It is entirely sufficient to claim that you're trying to learn language agnostic parameters/embeddings --  I'd be happy with that. But the paper goes further and claims to be learning a form of universal grammar. To justify this claims, it is not sufficient to show in the experiments that the new representations do better at sentiment and NLI. The authors must show that this captures the "innate" language learning abilities akin to human babies. While the paper aims to do some analysis in the discussion section, it is not unsatisfactory. As the paper says in the discussion section "From a machine learning perspective, we’re interested in extracting informative features and not necessarily a completely grammatical language model. That being said it is of interest to what extent language models capture grammar and furthermore the extent to which models trained toward the universal grammar objective learn grammar."

The problem is that simply comparing LM perplexities is not a solid test of whether this model has learned some form of universal grammar. First, this paper does not define a clear falsifiable hypothesis on the proof of learning universal grammar. One example of testing for learning grammar can be: does this model learn basic syntactic rules of a new language (e.g. as the authors suggested -- head-first or head-final syntax, or rules of conjugation)  with a small amount of data after being training a universal representation with n languages? There have been a series of recent papers on checking if Language Models have appropriately learned syntax. See e.g. Tal Linzen's work https://arxiv.org/pdf/1809.04179.pdf. Just to be clear I am not suggesting citing works in unpublished places but potentially using some of the tests suggested in these papers.

In conclusion, I think this work is useful but I also think it makes really grandiloquent claims without verifying them. That to me is a dangerous precedent.

---

> ### Author Response · Authors · 2018-11-06
> **RE: ICLR 2019 Conference Paper24 AnonReviewer1**
>
> Thank you for your review. I'd like to clear up that the claim of our paper was only to learn language agnostic representations, our goal was not to comment on the Universal Grammar hypothesis, but simply to take inspiration from it.
>
> In our introduction we explicitly state: "Our attempt to learn these [language agnostic universal] representations begins by taking inspiration from linguistics and formalizing UG as an optimization problem". In our discussion we comment on the learnability of grammar (POS argument) and state that we are simply "interested in extracting informative features and not necessarily a completely grammatical language model". Furthermore in our abstract we state the novelty of our paper is "present[ing] a method to decouple the language from the problem by learning language agnostic representations and therefore allowing training a model in one language and applying to a different one in a zero shot fashion." I don't believe we explictly make any claims about the Universal Grammar hypothesis.
>
> That being said, we would like to explicitly state in our paper that we are not making any claims about UG. Because of this we've included two statements, one in our introduction and the other in our discussion section explicitly stating that we are only taking inspiration from UG and not making any claims.
>
> The new addition in the last paragraph of our introduction states: "We do not make any claims about the Universal Grammar hypothesis, but simply take inspiration from it". The new addition in the first paragraph of the Discussion section states: "The goal of our paper is not to make a statement on the Universal Grammar hypothesis."
>
> Thank you once again for you review. Please let me know if you have further concerns.

---

> > ### Comment · AnonReviewer1 · 2018-11-17
> > **Thank you; a few more corrections**
> >
> > When you claim that you are formulating UG as an optimization problem, you're implicitly claiming you know how to convert UG into a mathematical form (which can then be solved.) If you claim you have formulated X as an optimization problem then it means two things:
> > 1. You know enough about X to define it's objective and constraints.
> > 2. If solve the optimization problem, the solution has the properties that you expect from X.
> >
> > Since you do not do so in this paper where X=Universal Grammar, you should rephrase all the places where you claim you have formulated UG as an optimization problem.
> >
> > E.g I'd rephrase the following lines in intro as
> > "Our attempt to learn these representations begins by taking inspiration from linguistics and formalizing UG as an optimization problem. We do not make any claims about the Universal Grammar hypothesis, but simply take inspiration from it."
> > as
> > "We take inspiration from the UG hypothesis and learn latent representations that are language agnostic. These representations allow us to solve NLP problems in new languages without any language-specific training data."
> >
> > Similarly, you need to revise the rest of sections.

---

> > > ### Author Response · Authors · 2018-11-24
> > > **RE: AnonReviewer1**
> > >
> > > Thank you for your reply. We see and agree with how stating "formulating UG as an optimization problem" can be problematic. We have updated our paper removing/augmenting this statement and statements similar to it.
> > >
> > > We hope that with this revision alongside the previous revision (explicitly stating that we are not making any claims) we have removed your concerns about us making any claims on UG.
> > >
> > > Please let us know if you have further concerns.
> > > Thank you.

---

### Meta-Review · Area_Chair1 · 2018-12-13
**Interesting technical work, but serious issues with framing**

**Confidence:** 4
**Recommendation:** Reject

**Metareview:**

This paper addresses a clear open problem in representation learning for language: the learning of language-agnostic representations for zero-shot cross-lingual transfer. All three reviewers agree that it makes some progress on that problem, and my understanding is that a straightforward presentation of these would likely have been accepted to this conference. However, there were serious issues with the framing and presentation of the paper.

One reviewer expressed serious concerns about clarity and detail, and two others expressed serious concerns about the paper's framing. I'm more worried about the framing issue: The paper opens with a sweeping discussion about the nature of language and universal grammar and, in the original version, also claims (in vague terms) to have made substantial progress on understanding the nature of language. The most problematic claims have since been removed, but the sweeping introduction remains, and it serves as the only introduction to the paper, leaving little discussion of the substantial points that the paper is trying to make.

I reluctantly have to recommend rejection. These problems should be fixable with a substantial re-write of the paper, but the reviewers were not satisfied with the progress made in that direction so far.